METHODS AND PROTOCOLS

# Comprehensive Wet-Bench and Bioinformatics Workflow for Complex Microbiota Using Oxford Nanopore Technologies

Christoph Ammer-Herrmenau,[a] Nina Pfisterer,[a] Tim van den Berg,[b] Ivana Gavrilova,[a] Ahmad Amanzada,[a] Shiv K. Singh,[a] Alaa Khalil,[a] Rohia Alili,[c] Eugeni Belda,[c] Karine Clement,[c] Ahmed Abd El Wahed,[d] ElSagad Eltayeb Gady,[e] Martin Haubrock,[b] Tim Beißbarth,[b] Volker Ellenrieder,[a] Albrecht Neesse[a]

[a]Department of Gastroenterology, Gastrointestinal Oncology and Endocrinology, University Medicine Goettingen, Goettingen, Germany
[b]Department Medical Bioinformatics, University Medicine Goettingen, Goettingen, Germany
[c]Faculty of Medicine, Sorbonne University, INSERM UMRS NutriOmics, Paris, France
[d]Institute of Animal Hygiene and Veterinary Public Health, University of Leipzig, Leipzig, Germany
[e]Faculty of Medicine, Al Neelain University/Ibn Sina Specialised Hospital, Khartoum, Sudan

**ABSTRACT** The advent of high-throughput sequencing techniques has recently provided an astonishing insight into the composition and function of the human microbiome. Next-generation sequencing (NGS) has become the gold standard for advanced microbiome analysis; however, 3rd generation real-time sequencing, such as Oxford Nanopore Technologies (ONT), enables rapid sequencing from several kilobases to >2 Mb with high resolution. Despite the wide availability and the enormous potential for clinical and translational applications, ONT is poorly standardized in terms of sampling and storage conditions, DNA extraction, library creation, and bioinformatic classification. Here, we present a comprehensive analysis pipeline with sampling, storage, DNA extraction, library preparation, and bioinformatic evaluation for complex microbiomes sequenced with ONT. Our findings from buccal and rectal swabs and DNA extraction experiments indicate that methods that were approved for NGS microbiome analysis cannot be simply adapted to ONT. We recommend using swabs and DNA extractions protocols with extended washing steps. Both 16S rRNA and metagenomic sequencing achieved reliable and reproducible results. Our benchmarking experiments reveal thresholds for analysis parameters that achieved excellent precision, recall, and area under the precision recall values and is superior to existing classifiers (Kraken2, Kaiju, and MetaMaps). Hence, our workflow provides an experimental and bioinformatic pipeline to perform a highly accurate analysis of complex microbial structures from buccal and rectal swabs.

**IMPORTANCE** Advanced microbiome analysis relies on sequencing of short DNA fragments from microorganisms like bacteria, fungi, and viruses. More recently, long fragment DNA sequencing of 3rd generation sequencing has gained increasing importance and can be rapidly conducted within a few hours due to its potential real-time sequencing. However, the analysis and correct identification of the microbiome relies on a multitude of factors, such as the method of sampling, DNA extraction, sequencing, and bioinformatic analysis. Scientists have used different protocols in the past that do not allow us to compare results across different studies and research fields. Here, we provide a comprehensive workflow from DNA extraction, sequencing, and bioinformatic workflow that allows rapid and accurate analysis of human buccal and rectal swabs with reproducible protocols. This workflow can be readily applied by many scientists from various research fields that aim to use long-fragment microbiome sequencing.

**KEYWORDS** 16S rRNA, bioinformatic workflow, buccal swab, DNA extraction, eNAT, eSwab, Kaiju, Kraken2, metagenomics, MetaMaps, Metapont, microbiome, Oxford Nanopore Technologies, ONT, rectal swab, sampling, sequencing, storage

Address correspondence to Albrecht Neesse, albrecht.neesse@med.uni-goettingen.de.

The advent of high-throughput sequencing techniques has recently provided an astonishing insight into the composition and function of the human microbiome. Disruption of the commensal microbiome, commonly referred to as dysbiosis, is linked to several diseases, such as obesity, diabetes, chronic inflammatory disorders, and cancer (1–6). Next-generation sequencing (NGS) contributed vastly to the accumulating evidence in microbiome research of the last decade and has transformed the microbial landscape by identifying an enormous quantity of unculturable microbes (7, 8). Due to its wide availability and decreasing sequencing costs, NGS has become the gold standard for advanced microbiome analysis (9). However, there are inherent limitations of this method. The length of reads does not exceed 300 bp in the case of Illumina MiSeq. Although long-fragment sequencing is technically possible, it is not widely used. Furthermore, it is limited to 10 kb, and the capability for microbiome analysis could not yet be shown (10). Thus, the resolution of marker gene sequencing for bacteria, 16S rRNA, only provides reliable information up to the genus level, whereas targeting of particular hypervariable regions (V1 to V9) influences the microbial composition (9, 11, 12). To gain insight into species and strain compositions by NGS, metagenomic sequencing is required. However, this approach is more expensive and difficult to conduct with low-biomass environments and/or samples with high host contamination, e.g., tissue samples (9). In these cases, microbiome profiling must rely on marker gene sequencing. The limitation of insufficient species resolution by 16S rRNA sequencing can be overcome by analyzing the whole 16S rRNA gene (13, 14). The 3rd generation of sequencing subsumes methods with two distinct benefits: long reads and the possibility of real-time sequencing. Oxford Nanopore Technologies (ONT) enable the sequencing from several kilobases to >2 Mb (15, 16). Thus, it becomes possible to assemble whole genomes of bacteria and eukaryotes or even complex repetitive parts of the human genome using a few or even one contig (15–18). Several groups showed the feasibility of ONT sequencing for microbiome analysis with a diverse set of samples, i.e., mock communities (14, 19–21), low-biomass microbial compositions such as dog skin or dust (14, 22, 23), antimicrobial resistances (24), and infective pathogens from clinical samples (25, 26).

To the best of our knowledge, few studies have focused on analyzing a complex microbiome structure with ONT (27). Hence, the impact of sampling and storage conditions, DNA extraction, bioinformatic classification, and library creation on the analysis of more advanced biological specimens have not been investigated in depth. In contrast to most previous clinical studies that used stool samples, we analyzed buccal and rectal swabs. Notably, stool samples have certain limitations for clinical studies. First, longitudinal studies are increasingly being performed (9), and stool may not be immediately available in the desired time window, e.g., in the outpatient department. Second, there are cohorts of patients from which a standardized acquisition of stool is not easily feasible, e.g., infants or incontinent geriatric patients. Furthermore, the different niches of gut microbes must be taken into account (28, 29). Previous studies highlighted a taxonomic difference between biopsy specimen, rectal swabs, and stool samples. To this end, 16S rRNA and metagenomic sequencing studies revealed that swabs harbor a microbiome comparable to that of stool samples but also contain bacteria that can be found more frequently in biopsy specimens than in stool samples (28, 30–32). Therefore, rectal swabs are considered to give insight into an intestinal niche that is located between the lumen (stool) and the gut mucosa (33). Importantly, bacteria adjacent to the mucosa are crucial for the interaction with the gut immune system (34). However, the required bowel preparation prior to colonoscopy and the invasiveness of the procedure are limitations. Hence, rectal swabs have become a convenient alternative, providing more information about mucosal adherent microbes than stool samples (33). Despite intraindividual differences regarding the collection method, previous studies give evidence of a preserved microbial signature enabling the differentiation between individuals (31, 35, 36). There is growing evidence that oral microbiota can be used as biomarkers for several diseases (1, 37, 38). Although different microbial

niches of the oral cavity were characterized recently (39, 40), there is still a significant lack of knowledge about the optimal sampling conditions. In past studies, patients were requested to refrain from food, drinks, and oral hygiene between 30 min and several hours (41–43). To our knowledge, no conclusive data exist clarifying which sampling point is ideal and what transient impact food and drinking have on the oral microbiome.

In particular, the bioinformatics remain challenging for ONT due to the combination of the high base call error rate of ~10% and long reads (44). Previous studies provided bioinformatics workflows (20, 25). However, these wrapper programs were designed for the identification of single infectious microorganisms. Here, we present a customizable workflow algorithm that can run on different systems due to its convenient docker format (Github; https://github.com/microbiome-gastro-UMG/MeTaPONT/). This program was able to reduce the false-positive rate more sufficiently than classifiers designed for NGS.

To address these caveats, we conducted comprehensive experiments establishing a reliable wet-bench and bioinformatic workflow for both 16S rRNA and metagenomic sequencing, including detailed protocols that are easily adaptable for other researchers.

## RESULTS

**Swab reliability and storage conditions.** Swabs are commonly used for analyzing different sites of human microbiota, and their practicability is widely proven (35, 39, 40). We hypothesized that the choice of the swab and its medium will have an impact on microbiome analysis.

eNAT and eSwab are known to preserve microbes reliably, and their feasibility for microbiome analysis was recently shown (45–47). For reasons of comparability, one stool batch from a volunteer was homogenized. The swabs were dipped in stool, and a small portion was directly transferred to lysis buffer (from the MagMAX microbiome ultra nucleic acid isolation kit). The samples were immediately extracted (day 0 [d0]) or stored for 3 or 7 days under different temperature conditions (Fig. 1a). Beta diversity showed a clustering of most storage conditions (Fig. 1b). The microbial composition at the species level revealed a wide overlap between the majority of samples (Fig. 1c). Two outliers significantly explained ~80% of the observed differences. These two samples were eSwabs stored at room temperature (RT) for 3 and 7 days. Facultative anaerobic genera, like *Escherichia coli*, *Citrobacter freundii*, *Enterobacter cloacae*, and *Klebsiella aerogenes*, were especially increased. Thus, we conclude that eSwabs must be frozen immediately after sampling. eNATs seem to preserve the microbial composition even if samples were stored at room temperature for 7 days. Despite the same procedure, eNAT swabs showed reduced DNA content after extraction and failed 16S PCR in some instances. A systematic comparison of DNA concentration revealed a significantly higher concentration in both specimens collected by eSwabs. The measured purity was also in parts significantly higher in eSwabs compared to eNATs (Fig. 1d). Therefore, eSwabs with immediate freezing were used for subsequent experiments.

**The quantity of stool has an impact on microbial composition.** Different microbes colonize the gut at different sites (28). Following this paradigm, it could be expected that rectal swabs with higher quantities of stool may harbor a different microbial composition. To this end, we defined three grades of stool contamination before allocating 12 rectal swabs from 2 independent people (grade 0, 3; grade +, 4; grade ++, 5) (Fig. 2a). Regarding the amount of microbial DNA compared to human DNA, there was a significant increase of microbial genes in the samples with a high quantity of stool (Fig. 2b). Despite low microbial DNA content in clean rectal swabs, metagenomic sequencing yielded a sufficient sequencing depth, with more than 7,200 microbial reads/sample, allowing profound microbial analysis. Interestingly, the microbial composition at the species level revealed, among others, a higher portion of *Corynebacterium* species, *Corynebacterium jeikeium*, and *Akkermansia muciniphila* in swabs defined as grade 0 (Fig. 2c). Accordingly, both genera (*Corynebacterium* and *Akkermansia*) were more frequently found adherent to the mucosa than in stool samples (32, 48). Besides these

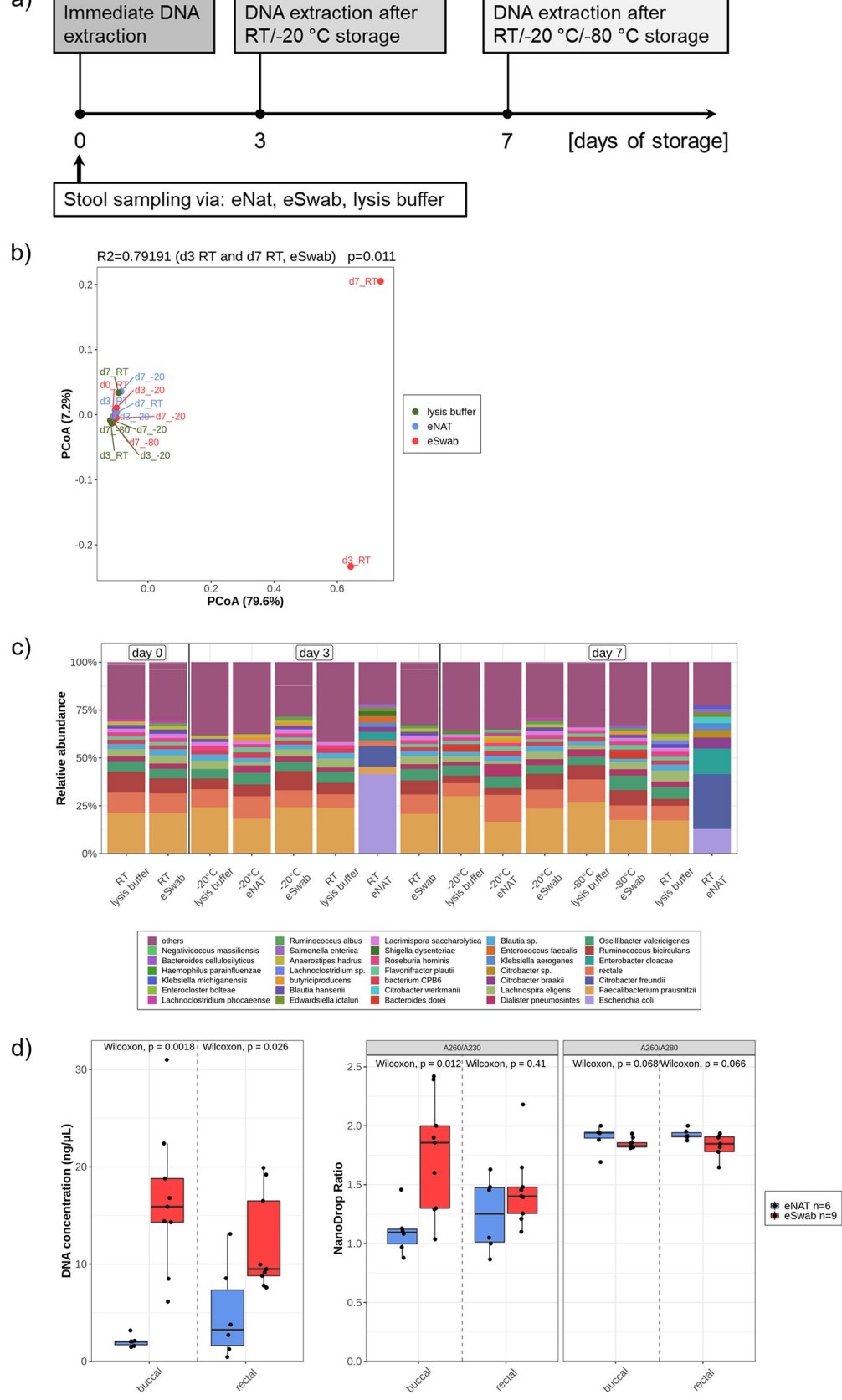

**FIG 1** Swab reliability and storage conditions. (a) Experimental design. (b) Beta diversity calculated with Bray-Curtis distance and ordinated with principal coordinate analysis. $R^2$ score and $P$ value explain the significant distance

minor differences, all swabs contained highly similar microbiota. All three mentioned species were also identified in more feculent swab samples with lower abundance (<2%) and were categorized in the green bar (other) (see Table S1 in the supplemental material). To further investigate whether highly feculent swabs are more similar to stool than to grade 0 swabs, 3 stool samples from consecutive days and 2 swabs within 7 h after bowel movement were collected from one individual. Indeed, the quantity of stool contamination has only a minor impact on microbial composition if swabs of different contaminations were compared to stool samples (Fig. S1).

**Impact of eating and drinking on the buccal microbiome.** Due to thermal and mechanical stimuli, it can be assumed that eating and drinking provoke a temporal alteration of the oral microbiome. Therefore, we aimed to investigate the impact of eating and drinking on the buccal microbiome (Fig. 3a). Unweighted UniFrac distance discovered a significant modification ($R^2 = 0.31$) of the buccal microbiome 5 and 30 min after eating (persons 1, 2, and 4) (Fig. 3b). Drinking water did not significantly affect the buccal bacterial composition. Importantly, despite the transient shift, the intraindividual microbial signature seemed to persist due to consistent personal clusters.

**Impact of DNA extraction protocol on DNA concentration, sequenced species, and read count for 16S rRNA and metagenomic sequencing.** There are several reports about the impact of DNA extraction methods on microbiome analysis (11, 46, 49–51). In order to standardize the microbiome analysis, a protocol by International Human Microbiome Standards (IHMS) is recommended for fecal samples (52). Therefore, we tested four different DNA extraction kits, of which three had shown reliable results for microbiome analysis from swabs (46) (Text S1). Two protocols achieved a significantly higher DNA concentration (measured by Qubit) in both buccal and rectal swabs compared to the other isolation methods. Invitrogen (IHMS) and Qiagen investigator kits (original) reproducibly yielded concentrations above the required threshold (>10 ng/$\mu$l, eluted in 50 $\mu$l) recommended for ONT metagenomic sequencing (dashed line) (Fig. 4a).

Despite some minor variability in $A_{260}/A_{280}$ ratio (Fig. S2a), most samples exceeded the required 1.8 value. However, remarkable divergences were observed regarding the NanoDrop $A_{260}/A_{230}$ ratio. Only the samples extracted by the Invitrogen (IHMS) protocol had values around the recommended $A_{260}/A_{230}$ ratio of 2 (Fig. S2b). To examine the reliability of extracted DNA methods on 16S rRNA and metagenomic sequencing approaches, at least four samples per protocol were sequenced. Using the rarefaction curve, a minimum required sequencing depth can be defined, where most samples reached a saturation. For 16S rRNA sequencing, a throughput of 250,000 sequences per sample was determined (Fig. 4b). The average alpha diversity did not show significant variations between the DNA isolation methods (Fig. 4c). However, most samples extracted by the Qiagen microbiome kit remained under this threshold of 250,000, and a tendency toward a lower alpha diversity was seen. Using unweighted UniFrac distance, which is recommended for rarefied samples (53), no DNA isolation protocol clustered significantly in both sample sites (Fig. S3).

Similar results were obtained by analyzing buccal and rectal swabs by metagenomics. Here, all samples isolated with the Qiagen microbiome failed to reach the threshold of 250,000 reads, and no differences regarding the alpha diversity were observed (Fig. 4d and e). Interestingly, there was a tendency where medians of samples isolated by IHMS protocols were higher regarding alpha diversity in metagenomic sequencing than their original counterparts. This might be explained by different read counts (sequencing depth). To prove whether the DNA extraction protocol has an impact on sequencing depth, five metagenomics experiments (10 to 12 samples per flow cell) were combined. Here, all samples were pooled equimolarly before flow cell loading. To

**FIG 1** Legend (Continued)

between d3_RT and d7_RT (both eSwab, outliers) and other samples. Distance calculation was performed at species level. (c) Microbial composition shows all species with >2% abundance, whereas the residual species were summarized as others (violet). All samples were normalized by prevalence filtering and rarefaction (10,000 reads/sample). (d) $n = 6$ buccal and rectal samples per swab were compared and purity was measured by NanoDrop.

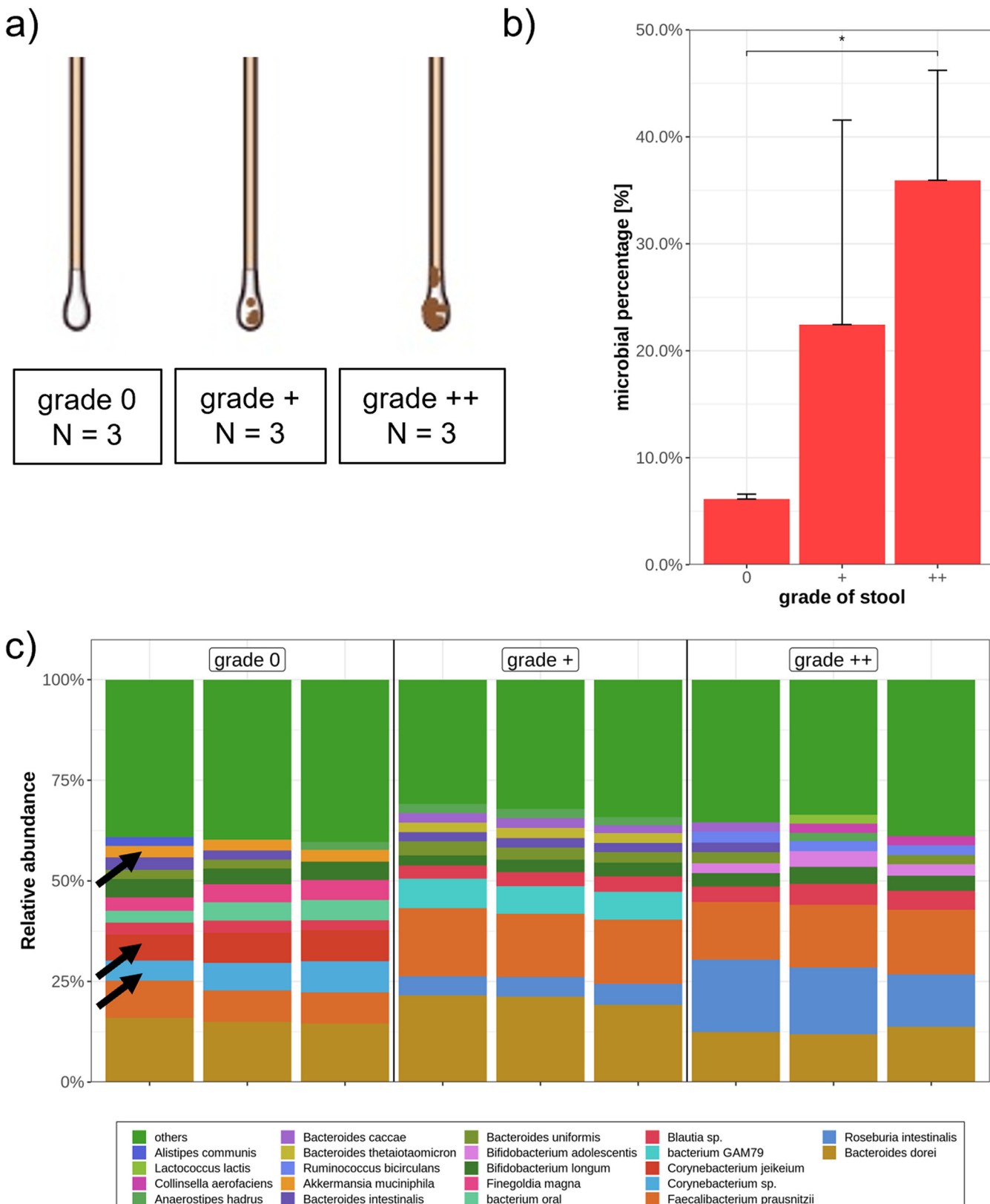

**FIG 2** Influence of quantity of stool on microbial DNA content and composition of rectal swabs. (a) Three different quantities of stool were compared. Twelve samples were collected and allocated by 2 people independently. (b) Average microbial DNA content of three defined grades of stool. Kruskal-Wallis and pairwise Wilcoxon rank test were performed with a *P* value of <0.05 (*). (c) Microbial composition of different stool quantities (3 samples per group) is presented at the species level, whereas all taxa under 2% are displayed as others (green). Black arrows indicate *Corynebacterium* species, *Corynebacterium jekeium*, and *Akkermansia muciniphila*, respectively. All samples were filtered for bacterial reads and rarefied to 7,200 reads/samples.

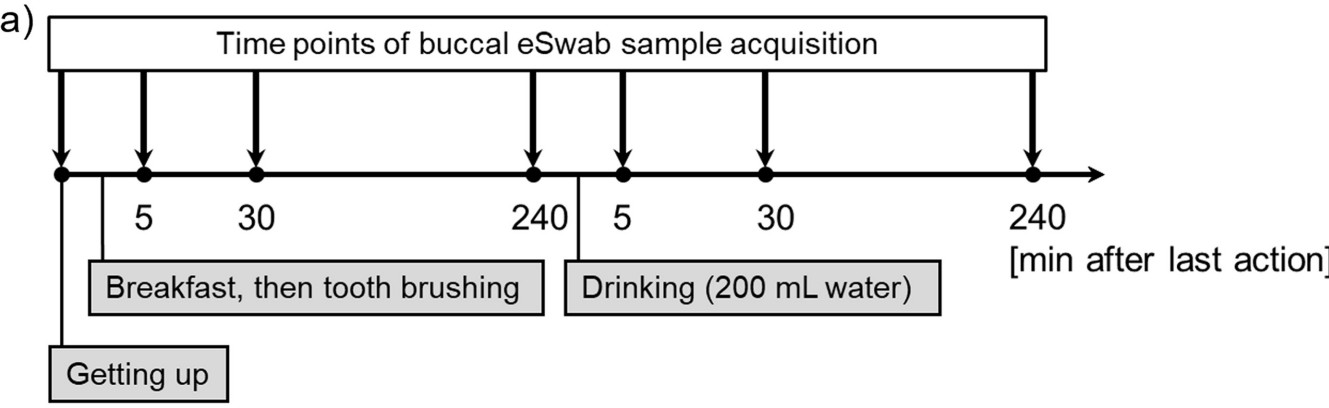

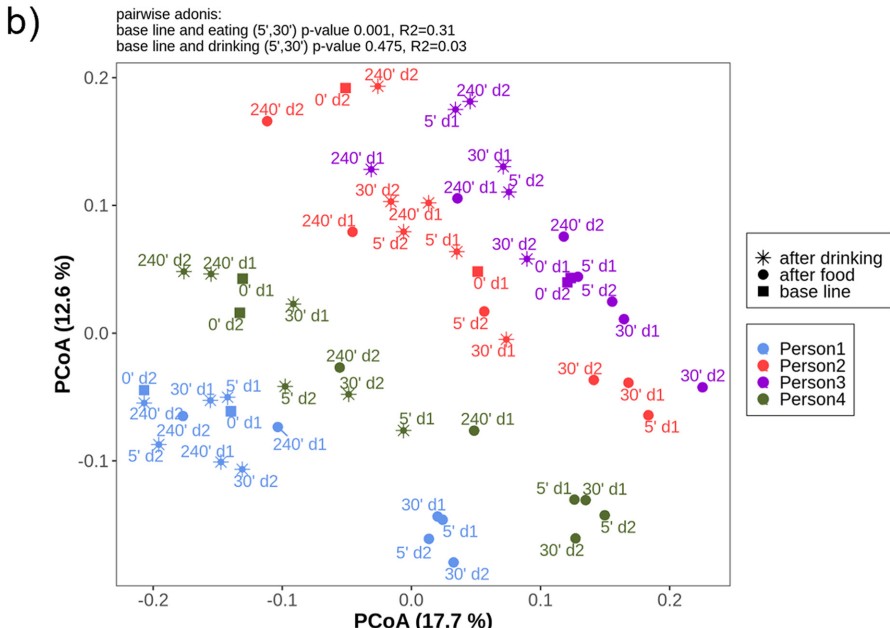

**FIG 3** Impact of eating and drinking on the buccal microbiome. (a) Experimental design. Four healthy nonvegetarian volunteers followed the protocol for 2 days. (b) Unweighted UniFrac distance ordinated with principal coordinate analysis showing the beta diversity at species level. Squares display the buccal swabs in the morning before eating (baseline), circles are the samples after eating (5 min, 30 min, and 240 min), whereas stars represent samples after drinking (5 min, 30 min, and 240 min). Samples were rarefied to 9,000 reads/sample.

avoid biases introduced by different flow cell pore counts, Qiagen IHMS was arbitrarily used as a reference. All read counts of the other methods were normalized to this protocol. Despite highly similar DNA input, a variety of sequencing depths per protocol was observed. Applied Biosystems and Qiagen investigator IHMS protocols tended to yield more sequences than the original protocols, whereas both Invitrogen protocols tended to yield the most profound throughput (Fig. 4f). Regarding the base call qscore, the protocols did not differ significantly and ranged between 11 and 12 (Fig. S4a and b). DNA extraction methods also had an impact on the average sequence length. Interestingly, Applied Biosystem and Qiagen investigator kits with the original protocols produced the longest fragments (Fig. S4c).

To summarize, we observed no biologically relevant differences regarding alpha and beta diversity and qscore from the base call. However, we found significant differences between extraction protocols regarding DNA concentration measured by Qubit and purity measured by NanoDrop, which has important implications for subsequent library preparation and sequencing throughput. Consequently, we used the PureLink microbiome DNA purification kit (Invitrogen), modified according to IHMS protocols, for further experiments.

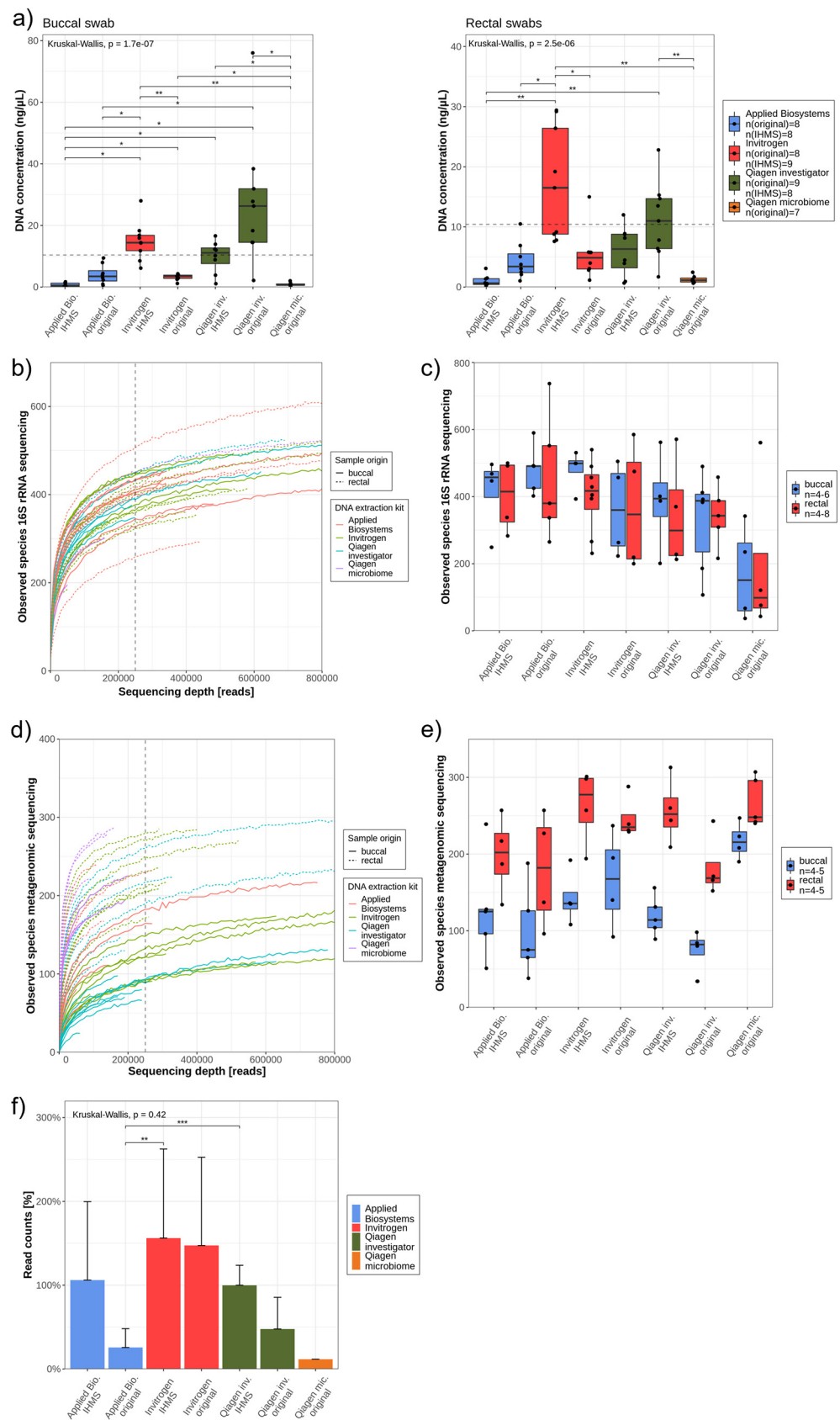

**FIG 4** Evaluation of DNA extraction protocols for 16S rRNA and metagenomic ONT sequencing. (a) Isolated DNA concentration from buccal und rectal swabs. Dashed lines at ~10 ng/μl represent the recommended DNA

mSystems®

**Classification of long noisy reads with Centrifuge.** The bioinformatic workflow applied for these experiments used three main programs (Fig. 5a): (i) Guppy for high-accuracy base calling; (ii) Centrifuge for classification; and (iii) Minimap2 as an alignment control (54, 55). Guppy was applied for base calling as it is currently the best-performing and the only officially supported base caller for ONT sequencing, outperforming traditional programs like Albacore, Flappie, and Scrappie (44). Centrifuge is used by the ONT-related platform Epi2me, which, among others, provides real-time 16S rRNA sequencing (14). This classifier has sensitive and fast annotation that do not require high computational capacity. Using 12 rectal swabs from the stool contamination experiment, a Venn diagram compared the read to taxonomic identification number (taxID) annotations between centrifuge alone (taking only the taxID with the highest quality score, which was the first in the row) and Centrifuge plus Minimap2 (Fig. 5b). More than one-quarter of sequence to taxID classification mismatched.

Low-quality sequences were excluded by defining cutoffs for (i) Centrifuge quality score and (ii) maximum number of annotations to a single read. Sanderson et al. postulated a threshold for a quality score of 150 (25). If this threshold was increased, a lot more sequences would be removed. With focus on the slope, defined by difference quotients ($g$), the increased threshold of the score did not influence the Minimap2 controlled sequences to such an extent as it did for the Centrifuge only classified reads (Fig. 5c). In other words, Minimap2 removed a high proportion of the reads, which will be excluded by a higher Centrifuge score. Therefore, the relatively low threshold of 150 proposed by Sanderson et al. was maintained.

For metagenomic sequencing, it is expected that highly conserved sequences or reads with redundant base sequences [like poly(A) fragments] will be obtained, resulting in a multitude of potential matches (for some sequences, more than 1,000 matches of different taxIDs per read were observed). To prove that these reads harbor low information and can be omitted to save computational time and to decrease the false-positive rate, we calculated a hit length/query length ratio. For most sequences, Centrifuge labeled fewer than 50 taxIDs, and the classification was based on an annotation length (hit length) of 40% of the total sequence length (query length) (Fig. 5d). The majority of sequences with more than 50 different taxID matches ranged below the first quartile of this ratio (Fig. 5e). Similar findings were seen regarding the Centrifuge quality score, which was mostly below the first quartile of all reads in the group with more than 50 matches (Fig. 5f). To investigate whether the underlying library is sufficient for microbiome studies, a mock community with 14 common gut bacteria, one archaeon, and 2 fungi was analyzed using metagenomic sequencing. We evaluated four different indices/libraries: 2 preformed indices supplied by Centrifuge ($p + h + v$ and $p$) as well as one library containing all NCBI Refseq complete genomes of bacteria, fungi, archaea, virus, and human and one including all incomplete and complete genomes in the NCBI nt database. These indices were analyzed by using three parameters: precision, recall, and area under precision recall curve (AUPR) (details in Text S1). These three parameters are commonly applied across benchmarking studies (56, 57). Preformatted indices ($pvh$ and $p$) contained only 11 and 13 species, respectively (Fig. 5g). Surprisingly, the comprehensive library including all NCBI Refseq complete genomes still failed to iden-

**FIG 4** Legend (Continued)
concentration for metagenomic experiments. $N = 7$ to 9 swabs were extracted per protocol and swab origin. (b) The rarefaction curve from 16S rRNA sequencing experiments displays continuous lines for buccal samples and dashed lines for rectal swabs. A sequencing depth cutoff at 250,000 was determined (black dashed line). (c) Alpha diversity of 16S rRNA sequenced samples was defined by observed species for buccal (blue boxplots) and rectal (red boxplots) swabs. At least $n = 4$ samples per biospecimen and DNA isolation protocol were sequenced. (d) The rarefaction curve was derived from buccal (continuous lines) and rectal (dashed lines) swabs, which were analyzed using a metagenomic approach. A sequencing depth cutoff at 250,000 was determined as a minimum read count (black dashed line). (e) Alpha diversity of samples, sequenced with a metagenomic approach, was defined by observed species for buccal (blue boxplots) and rectal (red boxplots) swabs. (f) Read counts in percentages were compared between different protocols after combining $n = 5$ different metagenomic sequencing experiments. Kruskal-Wallis and pairwise Wilcoxon rank test were applied to determine significance. *, $P < 0.05$; **, $P < 0.01$; ***, $P < 0.001$.

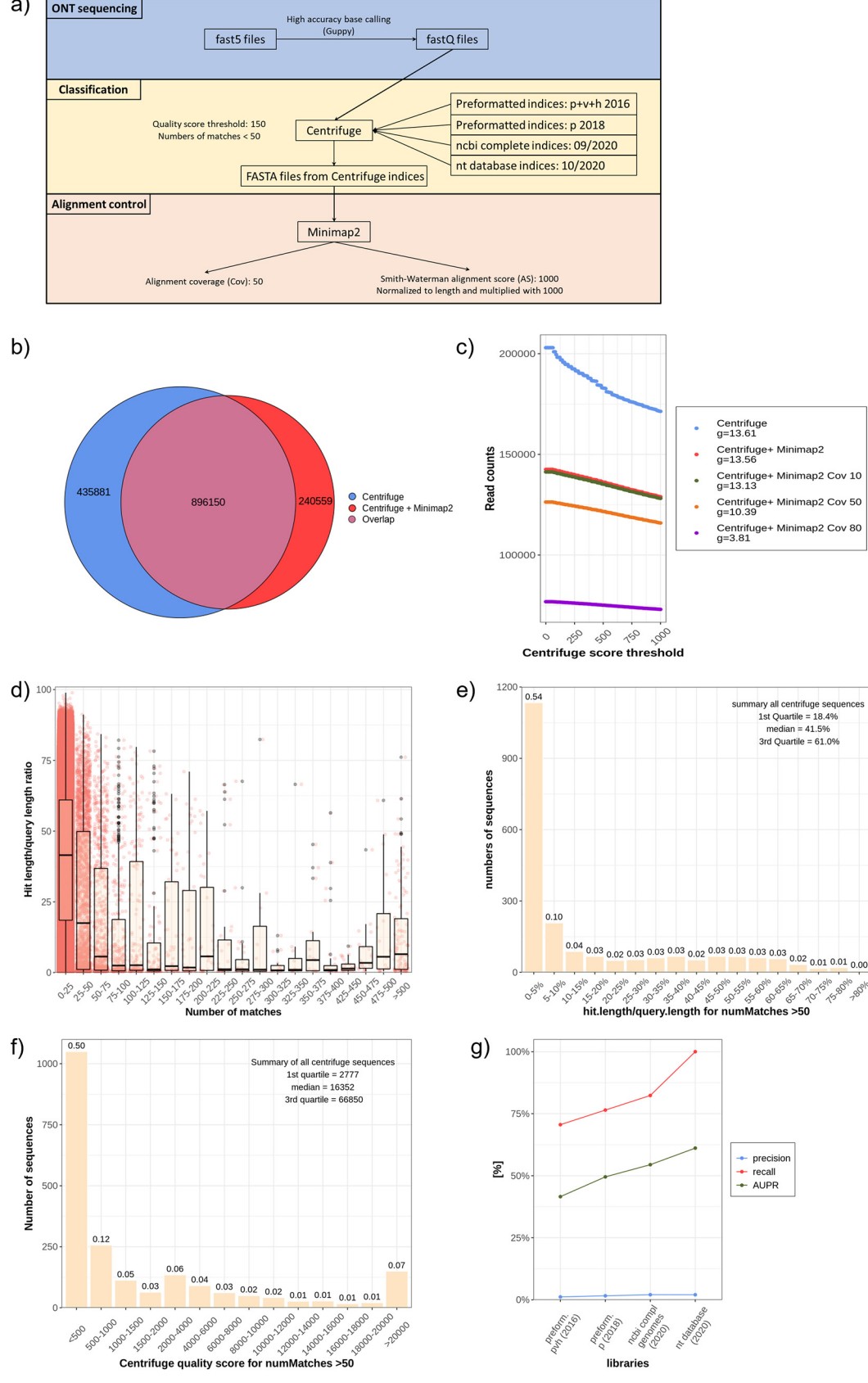

**FIG 5** Bioinformatic workflow and establishment of Centrifuge filter and library. (a) Overview of the bioinformatic pipeline with base calling (blue background), classification (beige background), and alignment control (red background). (b) Venn

tify 3 species from the mock community: *Prevotella corporis*, *Veillonella rogosa*, and *Candida albicans* (Table S2). A recall of 100% and high AUPR was only achieved by the index that includes all species from the NCBI BLAST nt database (Fig. 5g). The lowest precision was calculated for the preformatted index *pvh*.

**Evaluation of different Minimap2 parameters.** Centrifuge showed a convincing sensitivity, whereas the rate of false positives remained unacceptably high. Therefore, an alignment control by Minimap2 according to Sanderson et al. was established (25). To exclude low-quality reads and, thus, possible false-positive species, the alignment score (AS) and coverage (Cov) were further adjusted. To investigate which combination of these parameters kept all true-positive species and removed a maximum of false positives, a benchmarking was conducted by calculating precision, recall, and AUPR for each Cov and AS combination (Fig. 6a). An AS of 1,500 with a Cov of 50 obtained the highest precision and AUPR of all combinations with a recall of 100%. If the AS is increased to 2,000, the AUPR and precision values were still higher but also true-positive species were excluded.

To examine the impact of different Minimap2 coverages and alignment scores on the number of sequences and alpha diversity, 12 rectal swabs (metagenomic sequencing) and 8 (16S rRNA sequencing) samples were analyzed. As expected, more reads deriving from metagenomic sequencing were removed with an increased score and coverage (Fig. 6b). However, a higher AS influenced 16S rRNA sequences more than increased coverage. The same dynamics were observed for the alpha diversity (Fig. 6c). The precisions and AUPRs of both Cov 10 and 50 are comparable. To further investigate the impact of the coverage on microbial composition, 12 rectal swabs were benchmarked. The application of a higher Cov seems reasonable to filter out high-abundance, low-quality sequences (Fig. 6d). These reads were incorrectly annotated to environmental bacteria, i.e., *Mycobacterium branderi* and species C057 (*Nostoc*) or *Escherichia coli*, which are not usually observed with this high abundance or at all (black arrows in Fig. 6d).

The average read length was also affected by the coverage. Interestingly, a Cov of 80 significantly favored longer reads, whereas the average sequence length did not alter much between other AS and Cov thresholds (Fig. S5). To conclude, these benchmarking experiments and the established bioinformatic workflow that was tested on a mock community delivered highly accurate and reliable results. A diagram summarizes the optimized workflow (Fig. 7).

**Comparison of classifiers with simulated data.** There exists an increasing number of classifiers with different underlying algorithms that allow profound metagenomic analysis of complex microbial structures (56, 57). However, most of them were designed for NGS sequences, whereas some programs, like MetaMaps, were particularly developed for long reads with high error rates (58). Our presented workflow, Metapont, was compared to existing programs by using four simulated data sets. Deep learning allows us to simulate ONT sequencing with long reads and a realistic error rate (59). These simulated sequencing sets differ in their compositional complexity, containing 10 to 77 taxa (bacteria, fungi, and archaea) (Table S3). To this end, the *10-taxon-set* contained 8 pathogenic bacteria and 2 fungi. The *gut-set* was similar to the mock community used for evaluation of libraries and Minimap2 parameters. The *tumor-set* simulated a human tumor (diverse solid tumors) with 99.9% human DNA and 8

**FIG 5** Legend (Continued)
diagram with overlap between Centrifuge classification and Centrifuge + Minimap2 without any additional filter. (c) Influence of Centrifuge quality score on the number of sequences. *N* = 4 samples were combined after metagenomic sequencing and classified with Centrifuge and controlled with different Minimap2 filters (Cov). Differences of quotients (*g*) were calculated for each line. (d) *N* = 12 rectal samples were classified with a Centrifuge. The sequences were arranged to their number of matches to different taxIDs, and the classified length was divided through the total sequence length (hit length/query length ratio). Red dots represent a single read ID. All sequences with more than 50 different taxIDs were filtered, and their hit length/query length ratio (e) and qscore are presented (f). (g) A gut mock community was used to evaluate four different libraries of Centrifuge. They were compared with the following parameters: precision (blue line), area under the precision and recall curve (AUPR, green line), and recall (red line).

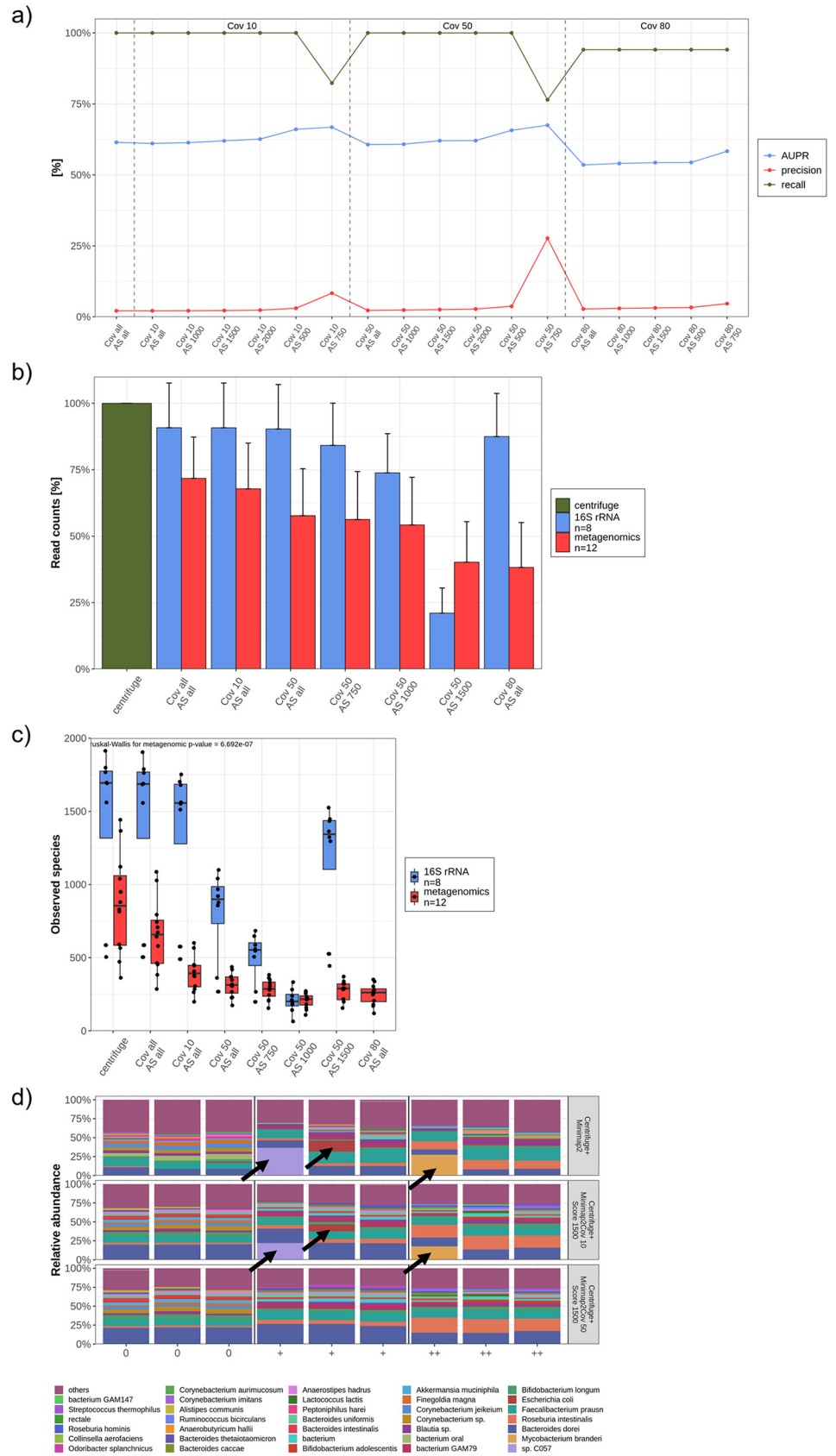

**FIG 6** Evaluation of different Minimap2 parameters. (a) A set of coverage (Cov) and alignment score (AS) thresholds were evaluated using the gut mock community. The thresholds were compared with the following

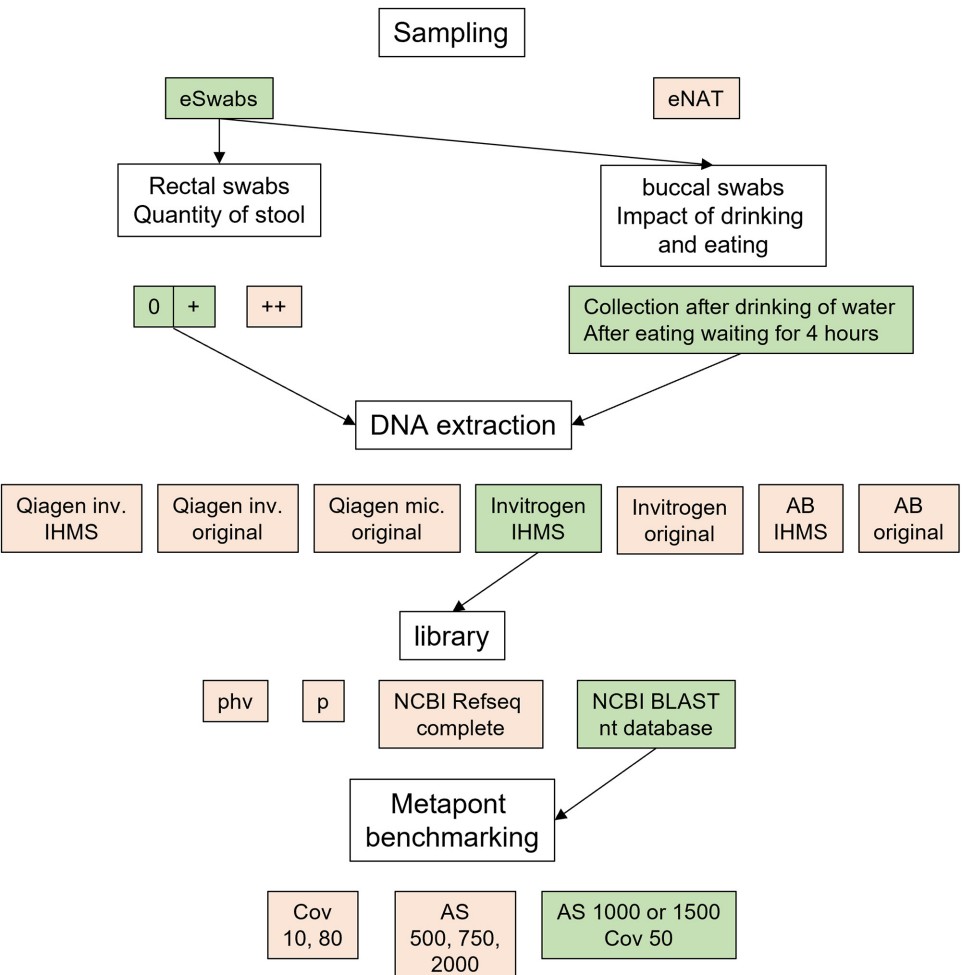

**FIG 7** Experimental overview. A schematic overview of the experiments. Parameters highlighted in green were set as the default and were used for the experiments if not mentioned otherwise. Parameters highlighted in red showed measurable disadvantages compared to the green ones.

different bacteria. This composition was based on the Nejman et al. intratumoral microbiome study (60). The last set with 77 taxa was adapted from the Dilthey et al. MetaMaps study (58). Metapont was compared with MetaMaps, Kraken2, and Kaiju (58, 61, 62). The last classifier is based on protein sequences, whereas the other ones are based on DNA sequences. Subsequently, the precision, recall, and AUPR were calculated. To avoid biases introduced by different libraries, comprehensive indices were built for Kaiju, Kraken2, and Metapont to recognize all taxa (recall of 100%). Unfortunately, currently there is no comprehensive library available for MetaMaps that includes all complete genomes of bacteria, fungi, and archaea (63). Therefore, this program yielded lower recalls except for the simulated data set, which was adapted from the programmers' study (58) (Fig. 8).

Notably, Metapont outperformed Kaiju and Kraken2 in all simulated sets in regard to precision, especially in the more complex microbiome sets (*tumor* and *metamaps*),

**FIG 6 Legend (Continued)**

parameters: precision (red line), area under the precision and recall curve (AUPR, blue line), and recall (green line). (b and c) The sequence count, in percentage (b), and the alpha diversity (observed species) (c) decreases with increasing coverage and scores for both metagenomic ($n = 12$) and 16S ($n = 8$) sequenced samples. Kruskal-Wallis test were calculated. (d) Microbial composition at the species level of $n = 9$ rectal swabs were displayed by applying three different filters (Minimap2 without any threshold, Minimap2 with a Cov of 10 and AS of 1,500, and Minimap2 with a Cov of 50 and AS of 1,500). Black arrows mark taxa that were filtered out by increased thresholds.

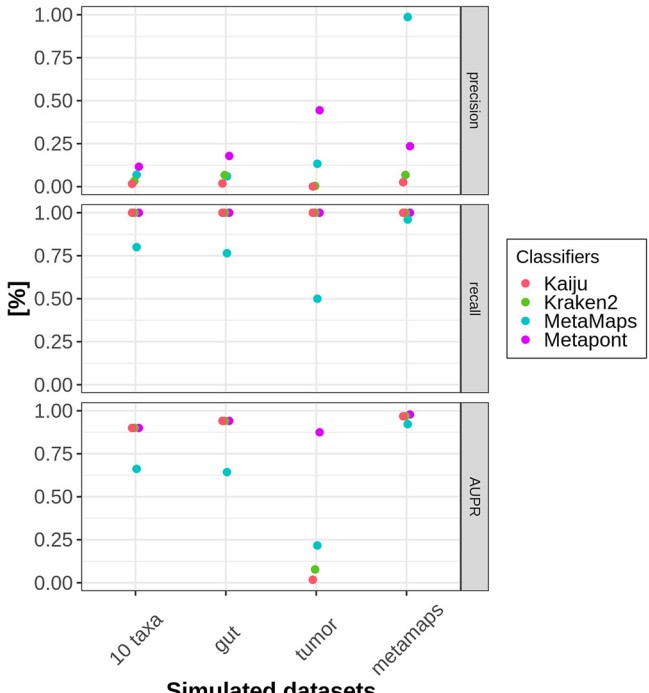

**FIG 8** Comparison of classifiers with simulated data. Four sets of simulated data were created and classified with four different programs: Kaiju (red line), Kraken2 (green line), MetaMaps (cyan line), and Metapont (purple line). Percentages of precision, recall, and AUPR gained by each classifier are displayed for each simulated data set.

whereas similar results for AUPR were yielded for these three classifiers. Given the link between the false-positive rate and the precision, these results demonstrate that the sequence of Centrifuge plus Minimap2 with adjusted AS and Cov can remove false-positive taxa most sufficiently in ONT sequencing experiments.

## DISCUSSION

Here, we present a benchmarking study for establishing a comprehensive workflow from sampling over DNA extraction and sequencing to bioinformatic analysis of microbiome data for ONT sequencing of complex microbiota. Although the validity of rectal swabs was shown several times before (33, 35, 36, 64, 65), there are several pitfalls for swab sampling. We could demonstrate that stool contamination has an impact on microbial DNA proportion and microbial composition. Therefore, to obtain reliable results, collecting rectal swabs with only minor stool contamination is recommended. Another drawback of using swabs instead of stool samples is a potential lack of biomass (31). However, our experiments proved that the amount of DNA isolated from swabs was reproducibly sufficient for complex microbiome analysis on the species level sequenced with 16S rRNA and metagenomic approaches.

Regarding the buccal swabs, our data revealed a transient alteration of the microbiome after food intake; nevertheless, an intraindividual microbial signature was preserved. Accordingly, the most reliable time point is the morning swab before tooth brushing and breakfast. If this time point is not feasible, a couple of hours need to pass between the last food intake and microbial sampling, whereas a small amount of drinking water will not disturb the microbial composition.

It is common knowledge that the method of DNA extraction strikingly influences microbiome analysis (50, 51). In general, IHMS modified protocols obtained more sequences than their original counterparts. A reason for the different sequence depths could be the portion of guanidine and other contaminants. The first one is known to interfere with the ONT library preparation and results in a low $A_{260}/A_{230}$ ratio by NanoDrop (66). These findings lead to the

conclusion that the IHMS protocol yielded higher throughput due to additional washing steps (see the supplemental material). The Invitrogen protocols (original and IHMS modified) have strikingly higher $A_{260}/A_{230}$ ratio values, which likely result in more sufficient sequencing depth due to the lower contamination (guanidine, residual salt, etc.) content of this kit.

There is growing evidence that ONT sequencing can perform accurate microbiome analysis (14, 19, 25, 67). In line with previous studies, we show that Centrifuge achieved a high sensitivity (19, 25). In contrast to Sanderson et al. (25) and Leidenfrost et al. (19), a more complicated mock community was chosen containing taxa that are not frequently implemented in preformatted libraries. To the best of our knowledge, all previous ONT microbiome studies classified with Centrifuge only used preformatted indices/libraries (*phv* or *p*). Interestingly, a database that was built according to official Centrifuge instructions, and that contained a big set of complete genomes from NCBI Refseq, failed to identify 3 species. One of them was *Candida albicans*, a well-known and important pathogen. Unfortunately, only 18 fungi were downloaded and included in this index. To this end, it is not advisable to use this library for complex microbiome analysis that aims to focus on fungi. Therefore, the most comprehensive database that detects all mock community species was used for all experiments if not stated otherwise. The drawback of a library that includes all known species is the high memory capacity and more time-consuming analyses.

Besides excellent AUPR and recall, the precision still remained low. This is explained by the structure of the analyzed mock community, which included extremely low-abundance species. Four species were below 0.1%, whereby the lowest-abundance species (*Clostridium perfringens*) accounted for only $1 \times 10^{-4}$%. Therefore, only all taxa below $1 \times 10^{-5}$% were filtered out by calculating the overall precision and recall. With this low prevalence filter, plenty of incorrectly annotated species were included in the analysis. If only the most abundant taxa were compared to the theoretical mock community composition, one species would be falsely classified, whereas the correct genus (*Veillonella*) would be annotated. We longitudinally investigated the reliability of 16S rRNA and metagenomic ONT sequencing for rectal and buccal swabs from the same individual. This longitudinal observation revealed a high consistency regarding microbial composition for both sequencing methods and biological specimens.

Furthermore, we simulated different sets of microbial ONT sequences and compared our workflow with three classifiers. MetaMaps was exclusively developed for long and noisy reads generated by ONT or PacBio (58). Currently, there is only one library available for this program containing >10,000 genomes. To this end, it performed best for the simulated data (*metamaps-set*), which was adapted from the community evaluated in the original study (58). However, it achieved insufficient AUPR and recall values, even in simulated sets with low complexity (*10-taxa*). This finding emphasized the need for a classification tool that analyzes complex microbiome structures sequenced by ONT with a comprehensive library. Metapont outperformed Kaiju and Kraken 2 regarding the precision in all simulated data sets. In other words, Metapont classified fewer false-positive reads than the other two conventional classifiers designed for NGS sequencing due to the adjusted Minimap2 coverage and the alignment score parameter.

**Conclusions.** Here, we present a comprehensive analysis pipeline with sampling, storage, DNA extraction, library preparation, and bioinformatic evaluation for complex microbiomes sequenced with ONT. Our findings from the swabs and DNA extraction experiments indicate that methods that were approved for NGS microbiome analysis cannot be simply adapted to ONT. We recommend using swabs and DNA extraction protocols with appropriate medium or extended washing steps. Both 16S rRNA and metagenomic sequencing achieved reliable and reproducible results. Still, the relatively high error rate of ONT sequences remains a bioinformatic challenge. Our benchmarking experiments reveal thresholds for analysis parameters that achieved excellent precision, recall, and AUPR values and is superior to existing classifiers. Hence, with the published bioinformatic pipeline, it is possible to achieve a highly accurate analysis of

complex microbial structures. This workflow can be easily downloaded as a docker and individually customized by other scientists.

## MATERIALS AND METHODS

**Swabs.** For the experiments designed to define the swab with the best reliability, Copan liquid Amies elution swabs (eSwab) containing 1 ml medium and eNAT (Copan) with 2 ml medium were evaluated. A stool sample from a nonvegetarian volunteer was homogenized with phosphate-buffered saline (PBS). Both swabs were dipped in the stool. Additionally, a small stool portion was directly mixed with lysis buffer. Subsequently, a set of three different samples were either directly extracted (d0), stored for 3 days (d3) at room temperature (RT) or at −20°C, or stored for 7 days (d7) at RT, −20°C or −80°C.

Buccal and rectal swabs for subsequent experiments were collected from a nonvegetarian volunteer using eSwabs. Rectal swabs were collected by one physician and were not self-administered. For this purpose, swabs were inserted 5 to 6 cm into the rectum and rotated 5 to 6 times. If not stated otherwise, rectal swabs had a quantity of stool ranging from 0 to + (Fig. 2a). For buccal swabs, both sides of the oral cavity were wiped thoroughly for at least 10 s. All samples were stored within 3 h at −80°C.

To analyze the impact of eating and drinking on the buccal microbiome, four nonvegetarian volunteers collected buccal swabs (eSwab) for 2 days according to the protocol (Fig. 3a). Text S1 in the supplemental material contains a detailed description. As a positive control, a gut mock community was purchased from ZymoBIOMICS gut microbiome standard (Zymo Research), extracted, and sequenced alongside swab samples.

All samples were derived from contributing scientists who are also listed as authors. The study was reviewed and approved by the Ethics Committee of the University Center Goettingen (no. 11/7/19).

**DNA extraction and purification.** Four DNA extraction kits were evaluated. To this end, two different protocols for three isolation kits were applied. First, the manufacturer's protocols, referred to as original, were used for MagMAX microbiome ultra nucleic acid isolation kit (Applied Biosystems), PureLink microbiome DNA purification kit (Invitrogen), QIAmp DNA investigator kit (Qiagen), and QIAmp DNA microbiome kit (Qiagen). Second, the protocols were modified according to the International Human Microbiome Standard (IHMS) (52). For the QIAmp DNA microbiome kit (Qiagen), a modification was not possible. If not stated otherwise, a PureLink microbiome DNA purification kit (Invitrogen) was used for DNA isolation. The detailed protocols are listed in Text S1. All samples were eluted with 50 $\mu$l elution buffer.

The purity and concentration of extracted DNA was measured using a Nanophotometer P330 (INTAS Göttingen) and Qubit3 (dsDNA HS assay kit; Thermo Fisher Scientific, Waltham, MA). Samples extracted with Qiagen kits or an Applied Biosystem kit were purified by a OneStep PCR inhibitor removal kit (Zymo Research) prior to sequencing. DNA isolated with Invitrogen only underwent a cleaning step if NanoDrop ratios ($A_{260}/A_{230}$ and $A_{260}/A_{280}$) were below 2 ($A_{260}/A_{230}$ ) or below 1.8 ($A_{260}/A_{280}$), respectively.

**Library preparation, sequencing, and base calling.** Extracted DNA samples were prepared for 16S rRNA gene sequencing using either SQK-16S024 (ONT) or SQK-RAB204 (ONT). For metagenomic sequencing, the ligation sequencing kit (SQK-LSK109) was applied in combination with Native Barcoding Expansion 1–12 (EXP-NBD104). Either 500 ng or the maximum amount of extracted DNA per sample was prepared for metagenomic sequencing. All samples were sequenced with MinION (ONT) or GridION (ONT) using R9.4 flow cells. The sequencing was controlled with the MinKNOW v. 20.06.4. The sequence duration ranged from 48 to 72 h depending on the throughput. Fast5 files were base called and demultiplexed using Guppy (ONT) version 4.0.15 running with GPU driver cuda with default parameters. All fastq files and corresponding metadata were uploaded in Qiita (study number 13720) (68) and https://github.com/microbiome-gastro-UMG/MeTaPONT/.

**Classification.** The bioinformatic workflow is summarized in Fig. 5a. Centrifuge (version 1.0.4-beta) was used for classification (55) and Minimap2 (version 2.17) for alignment control (54). A detailed description is provided in Text S1. A threshold filtering and the benchmarking were realized with a python script (version 3.7.7, pandas 1.1.3).

All listed tools were combined in a wrapper program. The classification analysis pipeline can be downloaded as a docker. Detailed instruction and download information can be found at Github at https://github.com/microbiome-gastro-UMG/MeTaPONT/.

Four different libraries/indices were tested with a gut mock community. Two were the preformatted indices, which can be downloaded from the official Centrifuge homepage (bacteria, archaea, viruses, human [compressed], 6 December 2016 [*phv*], and bacteria, archaea [compressed], 15 April 2018 [*p*]). The third was built on 10 September 2020 according to the official instructions of the Centrifuge manual containing all complete NCBI bacterial, fungi, archaea, human, and mouse RefSeq genomes (NCBI). Low-complexity regions were masked by NCBI-tool dustmasker (version 0.1.03). The fourth library comprises all genomes from the NCBI BLAST's nt database and was built on 15 October 2020.

**Simulated data.** Four sets of simulated data were built using DeepSimulator 1.5 (59). Fasta files were downloaded from NCBI, and steps for fast5 creation were conducted with standard parameters as instructed previously (69). A bash script for downloading, simulating, and allocation was written and is publicly available (https://github.com/microbiome-gastro-UMG/MeTaPONT/). Simulated data were classified with 4 programs: Kaiju, Kraken2, Metamaps, and Metapont. More details about the programs and libraries are provided in the supplemental material.

The *10-taxa-set* contains 8 bacteria and 2 fungi (abundance 10% per taxa). The *gut-set* included a similar microbial composition like mock gut community (with highly divergent abundances 14% to 0.0001%), 14 bacteria, 2 fungi, and 1 archaeon. Here, *Veillonella dispar* and *Prevotella intermedia* replaced the species from the same genus, which were found in the mock community. The *tumor-set* simulates a

human tumor microbiome. Eight bacteria constitute only 0.1% of the DNA, whereas the majority belongs to the host (*Homo sapiens*). Bacteria composition is similar to that of intratumoral microbiome (60). MetaMaps-set contained most species (77 bacteria) from the simulated data, which were analyzed by Dilthey et al. (58). Table S3 contains all simulated species names and abundances.

**Downstream analysis.** All downstream analysis was conducted by R (version 3.6.3). Operational taxonomic unit (OTU) tables were preprocessed by prevalence filtering and rarefaction. For filtering, PERFect was applied with the "simultaneous" approach (70). When rarefaction was performed, the figure legends explain the normalized OTU count per sample. Alpha- and beta-diversity, taxonomic composition, and rarefaction curve was calculated from a phyloseq object (packages: phyloseq 1.30.0, ape 5.4-1, tidyverse 1.3.0, ggplot2 3.3.2, vegan 2.5.6, pairwiseAdonis 0.0.1, data.table 1.13.0, gridExtra 2.3, ggpubr 0.4.0, car 3.0–10, reshape2 1.4.4, and rstatix 0.6.0). To generate the required taxon table, the program taxonkit (71) was applied and all taxID from NCBI were extracted in May 2020. ReadIDs, length, quality score, and barcode of each sequence were taken from sequencing_summary.txt, which was created during the Guppy base call.

**Statistics.** Before applying statistic calculations for $P$ values, normal distribution and homogeneity of variance was tested, performing Shapiro-Wilk test, plotting QQ-plot, and Levene test, respectively. If both conditions were given, $t$ test for two groups or analysis of variance (ANOVA) and *post hoc* Tukey test for data sets with more than two groups were performed. If no normality was tested, Wilcoxon rank test for two groups and Kruskal-Wallis with additional pairwise Wilcoxon test for more than two groups were performed. To determine the explained distance of beta diversity by certain factors, adonis and pairwise adonis tests were applied after checking normality and homogeneity of variance using betadisper and permutest. Beta diversities are ordinated with principal coordinate analysis (PCoA). For each plot, significances were visualized: *, $P < 0.05$; **, $P < 0.01$; ***, $P < 0.001$.

**Data availability.** The classification analysis pipeline can be downloaded as a docker. Detailed instructions and download information can be found at Github (https://github.com/microbiome-gastro-UMG/MeTaPONT).

## SUPPLEMENTAL MATERIAL

Supplemental material is available online only.
**TEXT S1**, DOCX file, 0.03 MB.
**FIG S1**, TIF file, 0.02 MB.
**FIG S2**, TIF file, 0.2 MB.
**FIG S3**, TIF file, 0.04 MB.
**FIG S4**, TIF file, 0.2 MB.
**FIG S5**, TIF file, 0.04 MB.
**TABLE S1**, PDF file, 0.3 MB.
**TABLE S2**, PDF file, 0.01 MB.
**TABLE S3**, PDF file, 0.5 MB.

## ACKNOWLEDGMENT

We have no conflict of interest to declare.

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
