## [Reviewer comments · mSystems]

Comprehensive wet-bench and bioinformatics workflow for complex microbiota using Oxford Nanopore Technologies

Christoph Ammer-Herrmenau, Nina Pfisterer, Tim van den Berg, Ivana Gavrilova, Ahmad Amanzada, Shiv Singh, Alaa Khalil, Rohia Alili, Eugeni Belda, Karine Clement, Ahmed abd el Wahed, ElSagad Gady, Martin Haubrock, Tim Beissbarth, Volker Ellenrieder, and Albrecht Neesse

Corresponding Author(s): Albrecht Neesse, University Medicine Goettingen

Review Timeline:

Submission Date:	June 16, 2021
Editorial Decision:	July 12, 2021
Revision Received:	July 18, 2021
Accepted:	July 20, 2021

Editor: Nicola Segata

Reviewer(s): The reviewers have opted to remain anonymous.

Transaction Report:

DOI: <https://doi.org/10.1128/mSystems.00750-21>

July 12, 2021

Prof. Albrecht Neesse
University Medicine Goettingen
Gastroenterology, Gastrointestinal Oncology and Endocrinology
Robert-Kochstr. 40
Goettingen
Germany

Re: mSystems00750-21 (Comprehensive wet-bench and bioinformatics workflow for complex microbiota using Oxford Nanopore Technologies)

Dear Prof. Albrecht Neesse:

Thank you for submitting your manuscript to mSystems. We have completed our review and I am pleased to inform you that, in principle, we expect to accept it for publication in mSystems. However, acceptance will not be final until you have adequately addressed the reviewer comments.

Preparing Revision Guidelines

For complete guidelines on revision requirements for your article type, please see the journal Article Types requirement at <https://journals.asm.org/journal/mSystems/article-types>. **Submissions of a paper that does not conform to mSystems guidelines will delay acceptance of your manuscript.**

Sincerely,

Nicola Segata

Editor, mSystems

Journals Department
Reviewer comments:

Reviewer #1 (Comments for the Author):

In this study classification of the microbiota from buccal and rectal swabs was performed using Oxford Nanopore Technology 16S and metagenomic sequencing comprehensively evaluating wet bench and bioinformatic work flows. Overall this study provides timely, valuable comparisons and a workflow for future studies. Thorough author revisions comprehensively addressed the major limitations mainly lack of comparison to other classification methods and higher classification resolution.

Nevertheless a few minor comments include

Line 35. Reorder bioinformatic classification and library creation

Line 53. Rephrase for clarity

Line 69. 'of the last years' rephrase for clarity

Line 106. Intestinal niche, please rephrase it is unclear which niche the authors are referring to small or large intestine, lumen or mucosa.

Reviewer #2 (Comments for the Author):

Reviewers Comments:

Authors addressed all points of this reviewer.

Background:

Background information is now more clear.

Results:

Added sections in results improved this part.

Classification of long noisy reads with centrifuge:

Line 238: "Guppy was applied for basecalling as it is currently the best performing and the only officially supported basecaller for ONT sequencing" Best performing in your setup and scientific question. Please rephrase / make this clear.

Comparison of Classifiers with simulated data:

Characteristics and composition of your simulated datasets show quite different performance. For the 10 taxa set results in Figure 8 look pretty similar for Kaiju, Kraken2 and Metapont. Can you explain the increase in precision of Metapont in comparison to Kaiju and Kraken2? Why is there a connecting line between the simulated datasets? These are independent.

Area Under Precision Recall: Kaiju and Kraken2 outperform Metapont for gut data set. Why?

Discussion:

Line 396: "The challenging high false positive rate is reduced most sufficiently by Metapont." Can you give an explanation, why this gets "reduced most sufficiently"?

Simulated data:

As mentioned before: Characteristics and composition of your simulated datasets show quite different performance. Have you tried other characteristics like 30%, 50% or 80% of one bacteria DNA to see if other classifiers outperform your approach in regards to precision for example? For rectal swabs this might be a good set and setup, but your manuscript states "Comprehensive wet-bench and bioinformatic workflow for complex microbiota using Oxford Nanopore Technologies".

Reviewer 1

Point 1. Line 35. Reorder bioinformatic classification and library creation

Response: We rephrased as recommended. (Line 34-35)

Point 2. Line 53. Rephrase for clarity

Response: We agree with the reviewer that this sentence needed to be more clarified. We rephrased this part and hope it is clearer now. (Line 51-52)

Point 3. Line 69. 'of the last years' rephrase for clarity

Response: We changed years to decade. NGS came up 2008 and since 2010 more and more microbiome research is performed using this technology. Therefore, we hope that this change brings more clarity. (Line 67)

Point 4. Line 106. Intestinal niche, please rephrase it is unclear which niche the authors are referring to small or large intestine, lumen or mucosa.

Response: Rectal swabs harbour microbes which were commonly found in stool but also bacteria which are more frequently observed mucosa adherent. Therefore, rectal swabs represent the microbiome of large intestines in general and a niche between lumen (stool) and mucosa. We added more information on this issue in the revised version of the manuscript. (Line 103)

Reviewer 2

Point 1. Line 238: "Guppy was applied for basecalling as it is currently the best performing and the only officially supported basecaller for ONT sequencing" Best performing in your setup and scientific question. Please rephrase / make this clear.

Response: We deleted the phrase "best performing" and added some more information.

Point 2. Comparison of Classifiers with simulated data:
Characteristics and composition of your simulated datasets show quite different performance. For the 10 taxa set results in Figure 8 look pretty similar for Kaiju, Kraken2 and Metapont. Can you explain the increase in precision of Metapont in comparison to Kaiju and Kraken2? Why is there a connecting line between the simulated datasets? These are independent. Area Under Precision Recall: Kaiju and Kraken2 outperform Metapont for gut data set. Why?

Response: As the reviewer correctly mentioned the characteristics and composition of the simulated datasets differ. To this end, the 10 taxa and gut community represents more simple compositions whereas the tumor and metapont sets are designed to be more complicated. Furthermore, the reviewer pointed out that recall and AUPR are highly similar between Kaiju, Kraken2 and Metapont. That is not surprising because we used a comprehensive library for all of the classifiers and we wanted to achieve a recall of 100%, meaning that all true positives are recognized by the classifier. Marked differences are observed regarding precision. That is due to a higher false positive rate annotated with Kraken2 and Kaiju compared to Metapont. With the help of the adjusted Minimap2 parameter false positively classified reads were removed. (Line 326-327)

We agree with the reviewer that the lines suggest a dependency which does not exist. Therefore, we only provide dots for figure 8.

Moreover, we like to thank the reviewer for the critical view to discover the discrepancy in the AUPR of Metapont in the gut dataset. We encountered a minor mistake in the analysis. After correction,

the AUPR is similar to Kraken2 and Kaiju, totally in line with the precision and recall values of this dataset.

Point 3. Line 396: "The challenging high false positive rate is reduced most sufficiently by Metapont." Can you give an explanation, why this gets "reduced most sufficiently"?

Response: We rephrased this sentence and hoped it clearer now. (Line 389-393)

Point 4. As mentioned before: Characteristics and composition of your simulated datasets show quite different performance. Have you tried other characteristics like 30%, 50% or 80% of one bacteria DNA to see if other classifiers outperform your approach in regards to precision for example? For rectal swabs this might be a good set and setup, but your manuscript states "Comprehensive wet-bench and bioinformatic workflow for complex microbiota using Oxford Nanopore Technologies".

Response: As stated in response to point 2, we designed different complex simulated sets on purpose to compare the classifier performance. Figure 8 shows that metapont outperformed the other classifier especially regarding the precision. That becomes more evident in more complex compositions as *tumor* and *metamaps* sets. Especially the *tumor* is more comparable to metagenomic sequencing as it comprises a lot of host DNA. Given the highly similar AUPR of Kaiju, Kraken2 and Metapont we do not expect different precision and recall performance if only parts of the simulated sets are analysed. Furthermore, DeepSimulator takes a whole fasta file of a species for simulating ONT sequencing. To this end, partial species genome simulating was not conducted.

July 20, 2021

Prof. Albrecht Neesse
University Medicine Goettingen
Gastroenterology, Gastrointestinal Oncology and Endocrinology
Robert-Kochstr. 40
Goettingen
Germany

Re: mSystems00750-21R1 (Comprehensive wet-bench and bioinformatics workflow for complex microbiota using Oxford Nanopore Technologies)

Dear Prof. Albrecht Neesse: Thanks for submitting a revised version of your paper. Your manuscript has been accepted, and I am forwarding it to the ASM Journals Department for publication. For your reference, ASM Journals' address is given below. Before it can be scheduled for publication, your manuscript will be checked by the mSystems senior production editor, Ellie Ghatineh, to make sure that all elements meet the technical requirements for publication. She will contact you if anything needs to be revised before copyediting and production can begin. Otherwise, you will be notified when your proofs are ready to be viewed.

As an open-access publication, mSystems receives no financial support from paid subscriptions and depends on authors' prompt payment of publication fees as soon as their articles are accepted. =

Publication Fees:

We recognize that the video files can become quite large, and so to avoid quality loss ASM